# Effect of Ionospheric Variability on the Electron Energy Spectrum estimated from Incoherent Scatter Radar Measurements

Oliver Stalder[1], Björn Gustavsson[1], and Ilkka Virtanen[2]

[1]Department of Physics and Technology, University of Tromsø, Norway
[2]Space Physics and Astronomy research unit, University of Oulu, Finland

**Correspondence:** Oliver Stalder (ostalder@outlook.com)

**Abstract.** The ion composition in the E-region is modified by auroral precipitation. This affects the inversion of electron density profiles from field-aligned incoherent scatter radar measurements to differential energy spectra of precipitating electrons. Here a fully dynamic ionospheric chemistry model (IonChem) is developed that integrates the coupled continuity equations for 6 ion and 9 neutral species, modeling the rapid ionospheric variability during active aurora. IonChem is used to produce accurate, time-dependent recombination rates for ELSPEC to improve the inversion of electron density profiles to primary electron energy spectra. The improvement of the dynamic recombination rates on the inversion is compared with static recombination rates from the International Reference Ionosphere (IRI) and the steady-state recombination rates from an ionospheric chemistry model, FlipChem. A systematic overestimation at high electron energies can be removed using a dynamic model. The comparison with FlipChem shows that short-timescale density variations are missed in a steady-state chemistry model.

## 1 Introduction

The aurora is a dynamical high-latitude phenomenon caused by magnetospheric electrons and protons with energies in the range of $keV$ precipitating into the ionosphere. Precipitation leads to rapidly varying ionization, excitation and heating over a large range of spatial scales, with higher primary electron energy causing more ionization at lower heights. Precipitation also impacts the E-region substantially by inducing compositional changes and increasing conductivities. Spatial and temporal changes in ionospheric conductivities affect currents and field-aligned potentials. Precipitation can dominate power deposition at small spatial and temporal scales (e.g., Palmroth et al., 2006), and, for all these reasons, plays an important role in magnetosphere-ionosphere (MI) coupling. The energy spectrum of primary electrons and its time variation make it possible to investigate the acceleration process in the magnetosphere causing aurora. Understanding the dynamics of MI coupling and the ionospheric response to rapid variation of precipitation are active research topics.

Ground-based measurements complement in-situ observations of electron precipitation. Satellites and rockets can measure the electron energy distribution directly, but their high velocities do not allow for extended measurements at a single location. Disentangling spatial from temporal variations can be challenging, and, for satellites, the resulting spatial resolution may be limited. Incoherent Scatter Radars (ISR) can follow the temporal evolution of auroral precipitation above the radar's field of view for extended time periods. On the other hand, ISR measurements for this purpose are restricted to the magnetic zenith

direction, and at high time resolutions one needs to adapt to high noise levels. Optical observations can complement ISR in the horizontal direction (e.g., Tuttle et al., 2014).

The electron energy spectrum can be estimated from the time-variation of E-region electron density profiles measured with ISR. The inversion starts from the electron continuity equation:

$$\frac{dn_e}{dt} + \nabla \cdot (n_e \boldsymbol{v}_e) = q_e - \bar{\alpha} n_e^2 \tag{1}$$

where $q_e$ is the ionization rate, $\bar{\alpha}$ the effective recombination rate and $\boldsymbol{v}_e$ is the electron drift velocity. Transport of energetic electrons along the magnetic field and ionization are governed by a set of coupled linear differential equations (e.g., Lummerzheim et al., 1989). The superposition principle applies for direct ionospheric responses such as ionization. Therefore it is possible to calculate the ionization profile as a matrix product with a discretized representation of the differential electron flux $\phi(E)$ at the top of the ionosphere:

$$q_e(z) = \boldsymbol{A} \cdot \phi(E) \tag{2}$$

where $\boldsymbol{A}$ is the ionization-profile matrix with the ionization rates at discrete energies $E$ and altitudes $z$ (e.g., Fang et al., 2010; Semeter and Kamalabadi, 2005; Sergienko and Ivanov, 1993; Rees, 1989). Quantifying the ionization rate profile $q_e(z)$ makes it possible to solve for the energy spectrum $\phi(E)$. For small-scale aurora, the convection of plasma in and out of the radar beam can be significant. However, since no full-profile multi-static velocity measurements have been available, the convective term

$\nabla \cdot (n_e \boldsymbol{v}_e)$ is usually neglected.

The first methods to perform this inversion estimated $q_e(z)$ from Eq. (1) assuming steady-state, i.e., that ion production and recombination are in balance at all times, e.g., UNTANGLE by Vondrak and Baron (1977) and CARD by Brekke et al. (1989). Kirkwood (1988) first considered non-steady-state conditions in the SPECTRUM algorithm, enabling reasonable estimates even when the electron precipitation varies on time-scales shorter than the recombination time, i.e., during auroral precipitation.

Semeter and Kamalabadi (2005) first formulated this as a general inverse problem and used the maximum entropy method to regularize the solution. However, these direct methods run into two problems: The amplification of measurement noise when $dn_e/dt$ and $n_e^2$ in Eq. (1) are taken directly from the measurements, and the ill-conditioned nature of the inverse problem from ionization profile to energy spectrum. Those are addressed with the ELSPEC method by Virtanen et al. (2018), where the electron density is modeled by integrating the continuity equation, using Eq. (2) for the production term. Thereby the explicit

calculation of $dn_e/dt$ is avoided. The inversion is recast into a non-linear minimization problem, selecting for the best energy spectrum that minimizes differences between measured and modeled electron density.

In this work we present a refined ELSPEC version, where a dynamic ionospheric composition model, IonChem, is added. IonChem integrates the continuity equation for 15 ionospheric species, capturing the full dynamics in composition even during rapidly varying auroral precipitation. This enables us to study the effects of variation in ionospheric composition and, in

consequence, the effective recombination rate on the estimation of primary electron spectra from ISR data. The method we present here aims to improve the electron energy spectra estimates and to study the effects of ionospheric variation.

## 2 Method

This section begins with a description of ELSPEC, the method used to estimate electron energy spectra from electron density profiles. The following section describes the ion chemistry model. In section 2.3 the coupling of IonChem into ELSPEC is explained. Next, the robustness of the technique under uncertain initial compositions is analyzed. In the last section a steady-state ion chemistry model, FlipChem, is introduced, which will be used for comparison.

### 2.1 ELSPEC

In this study, the ELSPEC algorithm (Virtanen et al., 2018) is used for inversion, extended by a robust statistics implementation [B. Gustavsson, unpublished]. ELSPEC estimates the primary electron differential number flux $\phi(E)$ $[\mathrm{m}^{-2}\mathrm{s}^{-1}\mathrm{eV}^{-1}]$ by searching for the parameterized spectrum that minimizes the corrected Akaike information criterion (cAIC). The cAIC is calculated as the residual sum-of-squares of the difference between observed $(n_e^o)$ and modeled $(n_e^m)$ electron density profiles during a time period, with a cost term for the number of free parameters $L$ that prevents overfitting for small sample sizes:

$$cAIC = \sum \frac{(n_e^o - n_e^m)^2}{\sigma_{n_e^o}^2} + \frac{2(L+1)(L+2)}{M-L} \tag{3}$$

with $\sigma_{n_e^o}^2$ being the variance in the observed electron density, and $M$ the number of measurements. The modeled electron density is obtained by integrating Eq. (1). Allowing for a variable number of free parameters, the cAIC selects for the best-fitting parametrization of the electron spectrum. The robust statistics implementation starts with a coarse time interval, assuming a constant energy spectrum over 128 electron density profiles, in this case corresponding to $56$ s. The time interval is recursively refined to the point where dividing an interval is not decreasing its cAIC anymore, or the time resolution imposed by the measurements is reached.

Typically an altitude-dependent, but time-constant recombination rate is used for the inversion. However, the effective recombination rate depends on the ion composition, in particular $O_2^+$ and $NO^+$, being the most abundant diatomic ions.

$$\bar{\alpha} = \alpha_{\mathrm{NO^+}} \frac{n_{\mathrm{NO^+}}}{n_e} + \alpha_{\mathrm{O_2^+}} \frac{n_{\mathrm{O_2^+}}}{n_e} \tag{4}$$

where $\alpha_{\mathrm{NO^+}}$ and $\alpha_{\mathrm{O_2^+}}$ are the recombination rates of $NO^+$ and $O_2^+$ with electrons, respectively. It has been shown that the ionospheric composition varies greatly during auroral precipitation (e.g., Jones and Rees, 1973; Zettergren et al., 2010). This impacts electron energy spectra (Virtanen et al., 2018). Newer versions of ELSPEC account for that to some extent, allowing $\bar{\alpha}$ to be calculated with FlipChem (Reimer et al., 2021), a steady-state model for ionospheric composition.

### 2.2 IonChem

To model the ionospheric response to the precipitation, the coupled continuity equations for electrons, ions and minor neutral species (e$^-$, H$^+$, N$^+$, O$^+$, N2$^+$, NO$^+$, O2$^+$, H, N(4S), N(2D), O(1D), O(1S), NO) are integrated in time:

$$\frac{dn_k}{dt} = q_k - l_k \tag{5}$$

where production $q_k$ and loss $l_k$ terms for the ion species $k$ are of the form

$$q_k = q_{A,k} + \sum_{(i,j) \to k} \alpha_{ij} n_i n_j \tag{6}$$

$$l_k = n_k \sum_i \alpha_{ik} n_i \tag{7}$$

summed over all reactions relevant for the species $k$. Convection, as for electrons, is neglected. Table A1 shows the reactions,
their rates $\alpha_{ij}$ and yields taken into account. The reaction rates are generally temperature dependent. The ion chemistry is
driven by impact ionization $q_{A,k}$ of the major neutral species (Rees, 1989):

$$q_{A,O^+} = q_e \frac{0.56 \, n_O}{0.92 \, n_{N_2} + n_{O_2} + 0.56 \, n_O} \tag{8}$$

$$q_{A,N_2^+} = q_e \frac{0.92 \, n_{N_2}}{0.92 \, n_{N_2} + n_{O_2} + 0.56 \, n_O} \tag{9}$$

$$q_{A,O_2^+} = q_e \frac{n_{O_2}}{0.92 \, n_{N_2} + n_{O_2} + 0.56 \, n_O} \tag{10}$$

Density profiles of the major species ($O_2$, $N_2$, and O) are assumed to be unaffected by the precipitation. The initial composition
is taken from the NRLMSIS2.1 and IRI-2012 models (Emmert et al., 2022; Bilitza et al., 2014), or the FlipChem model.

The reaction rates and densities span over a wide range of magnitudes. This can lead numerical ODE solvers to choose
excessively small integration steps. The integration therefore may take a long time, or fails to integrate the system of coupled,
non-linear, ordinary differential equations described in Eq. (5) (Nikolaeva et al., 2021). This is called a stiff problem, and a
stiff solver may be used to address it. Here we use the BDF solver from the Python SciPy package.

## 2.3 Coupling ELSPEC and IonChem

ELSPEC solves an optimization problem for every time interval over which the energy spectrum is assumed to be fixed,
evaluating the cAIC many times to find the best fitting electron spectrum. Ideally, the electron continuity equation should be
integrated together with the other continuity equations, as they are coupled. However, this would be computationally expensive.
Instead, an iterative approach was adopted, illustrated in Figure 1. ELSPEC only integrates the electron continuity equation,
assuming fixed recombination rates over the duration of the measurement's time resolution. The electron continuity equation
is thereby effectively decoupled from the ionospheric chemistry, simplifying the problem substantially. The resulting energy
spectra $\phi^i(E)$ of the $i^{th}$ iteration are used to calculate the ionization rates $q^i_{A,k}$ in altitude and time. IonChem then uses these
ionization rates to calculate the evolution of the minor species $n^i_k$ and effective recombination rate $\bar{\alpha}^i$ for every measurement in
altitude and time. The updated recombination rates are then used by ELSPEC in the $(i+1)^{th}$ iteration to find the optimal energy
spectrum $\phi^{i+1}(E)$. When a repeated iteration over ELSPEC and IonChem converges, i.e., $\bar{\alpha}^i \approx \bar{\alpha}^{i+1}$ and $\phi^i(E) \approx \phi^{i+1}(E)$, a
solution is found.

The convergence of this method is shown in Figure 2 for a test case. Over a few iterations, the effective recombination rate is
converging to negligibly small deviations between iterations, measured in relative variation between iterations $(\bar{\alpha}_{i-1} - \bar{\alpha}_i)/\bar{\alpha}_i$.
A relative accuracy of $10^{-7}$ is achieved, corresponding to the solver's relative accuracy setting.

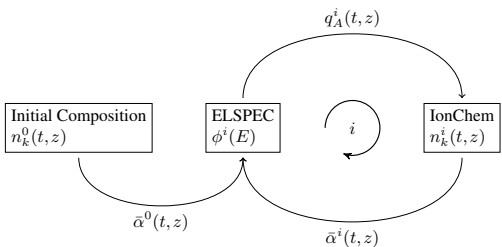

**Figure 1.** The iterative approach to resolve the computational challenge of pairing ELSPEC and IonChem is shown. ELSPEC is initialized with a model composition, and finds the optimal ionization rates. These are used by IonChem to find the ion densities, which then serve as the next model composition for ELSPEC. After several iterations $i$, the result is expected to converge.

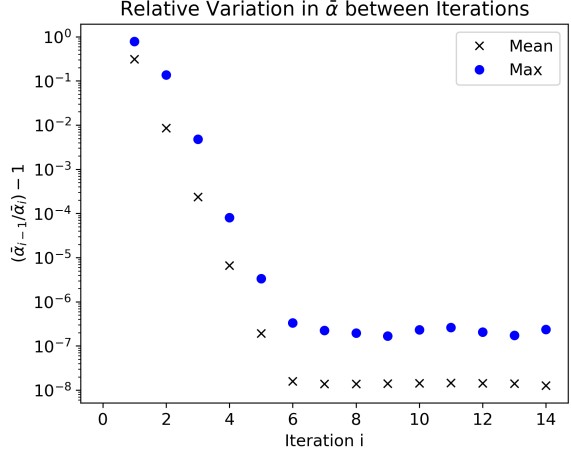

**Figure 2.** The relative variation in effective recombination rate between iterations $(\bar{\alpha}_{i-1} - \bar{\alpha}_i)/\bar{\alpha}_i$ shows clear convergence, both in the mean over all altitude and time bins, as well as the maximum value in all bins.

### 2.4 Initial Composition

The system of coupled continuity equations is non-linear and can, in principle, be sensitive to the initial conditions. In addition, the ionospheric composition can change significantly during auroral precipitation. The IRI model does not account for local auroral precipitation, it represents quiet-time conditions. IonChem therefore adds 30 minutes in front of the data set, during which a constant ionization rate is assumed, equal to that of the first ionization rate determined by ELSPEC. The model ionosphere thereby approaches a steady-state consistent with the prevalent precipitation. Figure 3 shows the $NO^+$ density and ionization rates at $96$ km altitude for the first iteration of ELSPEC and IonChem. ELSPEC starts at $t = 0$ s, integrating the

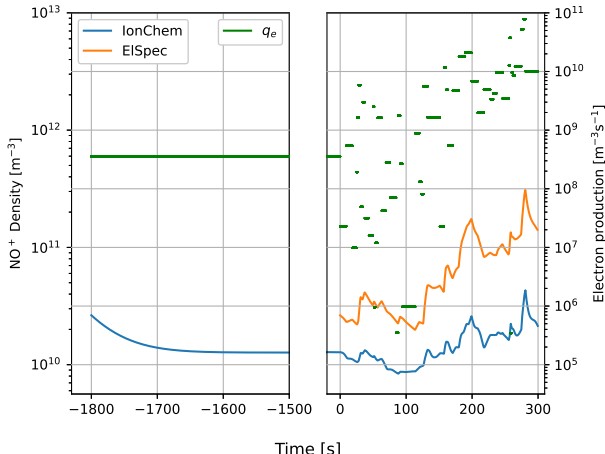

**Figure 3.** The effect of the 30 minute settling phase is shown on the example of $NO^+$ at 96 km altitude. The orange line shows the $NO^+$ density model used in ELSPEC, and the blue line shows the first iteration of IonChem. The green line shows the ionization rate. Before $t = 0$ s, the ionization rate is held constant, leading the IonChem composition to approach a steady-state, before the ionization rate is allowed to vary again.

continuity equation to determine the electron density. For the first iteration, ELSPEC initializes with IRI composition, i.e., the $NO^+$ density scales linearly with the electron density. The IonChem model starts at $t = -1800$ s, keeping the ionization rate
constant for the first 30 minutes. The $NO^+$ density plateaus at a steady-state until $t = 0$ s, when the ionization rate varies again. For some species, such as NO or N(4S), 30 minutes will not be enough to reach steady-state. It is preferable to have a good estimate of the initial conditions, but in the limited time between auroral events, some species will never reach steady state. In a test for the implementation, IonChem was initialized with constant production rates and temperature and run until steady-state. A steady-state at constant production and without photodissociation was approached after roughly 50 hours, consistent
with earlier studies (Roble and Rees, 1977; Bailey et al., 2002). Furthermore, the rapid variations during auroral precipitation will cause the densities to deviate further from steady-state. The exact state of the ionosphere is not known and will not be at steady-state, making it difficult to find the exact initial conditions for IonChem. A limited time to approach a steady-state allows the densities to reach reasonable levels, placing them in the right range with the prevalent conditions.

The robustness of this approach with regard to uncertain initial conditions is tested by running IonChem with different initial
compositions, specified in Table 1. These test cases should cover a fairly wide range of possible initial conditions, while still being reasonably close to reality. Cases 1-5 change the initial ion composition while leaving the plasma density unchanged. Cases 6-8 also change the plasma density, purposely generating a state rather far from steady-state. The effect of the NO initial density is investigated in cases 9-11.

All initial compositions produce very similar final ion composition variations. Figure 4 shows the $NO^+$, $O_2^+$ and $O^+$ densities
for all test cases. All lines are coinciding, showing that IonChem results are robust even when the initial ion composition unknown. Uncertain initial conditions do not seem to severely impact the solutions of this non-linear problem.

**Table 1.** Start conditions

| Case | Initial Composition | | Comments |
|------|--------------------|---|----------|
| 1 | IRI-2012, NRLMSIS2.1 composition | | Reference for comparison. |
| 2 | Swapping $O_2^+$ and $NO^+$ densities compared to IRI composition. | | During intense precipitation, the directly produced $O_2^+$ densities may surpass $NO^+$ levels. |
| 3 | $n_{O_2^+} = n_{NO^+} = 1/2\, n_e$ | | Starting with $O_2^+$ and $NO^+$ ions in equal amounts at all altitudes |
| 4 | $n_{O^+}$ | $= 2\, n_{O^+,IRI}$ | An increased fraction of $O^+$ is assumed due to ongoing precipitation. The densities of the other ions are re-scaled to maintain the plasma density. |
|   | $\sum n_{ions}$ | $= n_e$ | |
| 5 | $n_{O_2^+}$ | $= 1.5\, n_{O_2^+,IRI}$ | An increased fraction of $O_2^+$ is assumed due to ongoing precipitation. The densities of the other ions are re-scaled to maintain the plasma density. |
|   | $\sum n_{ions}$ | $= n_e$ | |
| 6 | $n_{O_2^+}$ | $= 2 n_{O_2^+,IRI}$ | An increased $O_2^+$ density is assumed due to ongoing precipitation. The plasma density is increased by the additional ion density. |
|   | $n_e$ | $= \sum n_{ions}$ | |
| 7 | $n_i$ | $= 2\, n_{i,IRI}$ | All ion densities are increased. The plasma density is increased by the additional ion density. |
|   | $n_e$ | $= \sum n_{ions}$ | |
| 8 | $n_{N_2^+}$ | $= 0.002\, n_{O^+}$ | The starting density of $n_{N_2^+}$ is set to be non-zero. The plasma density is increased by the additional ion density. |
|   | $n_e$ | $= \sum n_{ions}$ | |
| 9 | $n_{NO}$ | $= 10\, n_{NO,NRLMSIS2.1}$ | Increased NO density is assumed due to ongoing precipitation. |
| 10 | $n_{NO}$ | $= 100\, n_{NO,NRLMSIS2.1}$ | Increased NO density is assumed due to ongoing precipitation. |
| 11 | $n_{NO}$ | $= 0$ | No NO density for comparison with earlier MSIS models. |

## 2.5 FlipChem

To compare the effect of a dynamic ion chemistry model on the inversion, we use a steady-state ionospheric chemistry model FlipChem (Reimer et al., 2021) for reference. FlipChem is a Python interface to the Ion Density Calculator, a steady-state model for ionospheric composition (Richards et al., 2010; Richards and Voglozin, 2011). It uses electron density profiles to calculate production profiles and ion densities in the lower ionosphere, under the assumption that production and losses are balanced at any time (meaning the species are in "steady-state"). It is primarily intended to calculate density variations due to slowly varying photoionisation.

FlipChem allows the ion composition to adjust to slow variations in ionization, a non-linear process due to the chemical reactions. In contrast, using IRI composition corresponds to a linear scaling of quiet-time ion densities. Still, rapid variations in ionization may cause an imbalance of production and loss terms, causing the short-timescale dynamics to be missed under steady-state assumptions.

FlipChem is already implemented in newer versions of ELSPEC. Usually, it is used to calculate the ionospheric composition from the measured electron density in a pre-processing step.

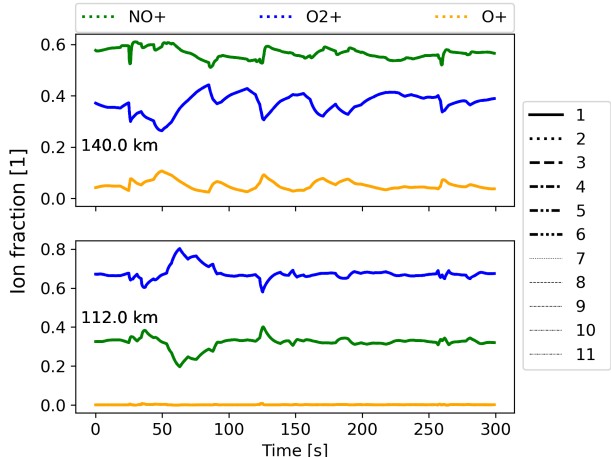

**Figure 4.** The integrated ion densities starting from different initial compositions are shown. The ion fractions $n_k/n_e$ of the most important ions $NO^+$, $O_2^+$ and $O^+$ are plotted for 2 different altitudes. The densities coincide for all runs, showing that all initial conditions produce the same densities.

155 Here, we use FlipChem to investigate the differences between a steady-state and a fully dynamic ion chemistry. To avoid propagating noise from the raw electron density measurements into the ion densities, in this work we first run ELSPEC with IRI ion composition to produce smooth electron density profiles, which are used to run FlipChem. The resulting, smooth FlipChem ion densities replace the IRI model in a second run of ELSPEC. This corresponds to a one-step iteration with the IonChem method presented above. This way, no noise in the electron density measurements affects either IonChem or FlipChem directly,
160 making for a fairer comparison between the models.

## 3 Results

To test this method and analyze the impact of ion composition variations on the electron energy spectra, we look at an event with rapid electron density variations in the E-region. Fig. 5 shows enhanced electron densities when several auroral arcs passed over the radar at about $50$ s and $125$ s. The data was recorded on the 12[th] of December 2006, 19:30-19:35 UT with the EISCAT
165 UHF radar in Tromsø (Dahlgren et al., 2011). GUISDAP (Lehtinen and Huuskonen, 1996) is used to evaluate EISCAT lag profile data. A time resolution of $0.44$ s in electron density is achieved, with the ISR operating in the arc1 experiment mode. At this time resolution the raw back-scattered power is used as a measure for electron density, under the assumption that electron and ion temperature are equal. The reaction rates are calculated using ion and electron temperatures estimated from the ion line spectra, integrated for $4$ s to achieve an acceptable noise level. They are interpolated to match the time resolution of the
170 electron density.

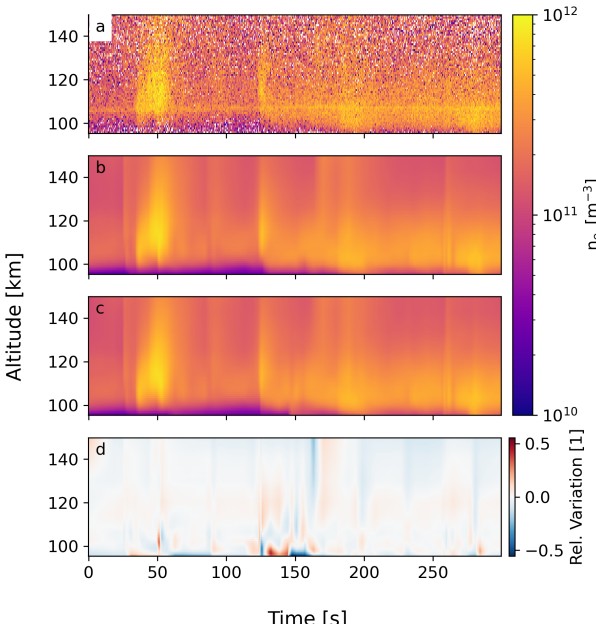

**Figure 5.** The measured electron density (a) is compared to the ELSPEC electron density models, using IRI composition (b) and IonChem composition (c). The measurements are reproduced well in both models. The relative variation $(n_{e,IRI} - n_{e,IC})/n_{e,IC}$ between the models in panel (d) shows that differences are small, with the biggest deviations occurring where the electron density is small.

The measured electron density as well as the ELSPEC electron density models based on IRI and on IonChem composition are shown in Figure 5, along with the relative variation $(n_{e,IRI} - n_{e,IC})/n_{e,IC}$. ELSPEC produces very similar electron densities with both composition models, modeling the measurements well in both cases. The differential energy flux $I_E = E\phi(E)$ produced using the IRI and IonChem models is shown in Figure 6, along with the difference $\Delta I_E = I_{E,IRI} - I_{E,IC}$ between the two. For three different slices in time, the results are also shown as a line plot, with the time marked in dashed lines in panel (c). The absolute difference (c) shows systematic negative values at the high energy tail, around 10 keV. The differences are substantial as can be seen in the line plots, with a relative correction of about 50 %. Therefore, using IonChem composition, ELSPEC produces a lower flux at high energies for all time intervals. This systematic difference is explained by the ratio $r = n_{O_2^+}/n_{NO^+}$, shown in Figure 7. IonChem composition has a higher ratio than IRI, especially towards the lower E-region. This is expected, since IRI does not take auroral precipitation into account and therefore represents quiet conditions where $NO^+$ is more abundant. During auroral precipitation, however, the direct production of $O_2^+$ makes it the dominant species at lower altitudes. Due to the lower recombination rate of $O_2^+$, a lower flux of primary electrons is sufficient to produce the same electron density. Using the IRI composition, representing the ionosphere at a quiet state, therefore leads to a systematic bias in the energy distribution, overestimating the energy flux at higher energies.

Another comparison is made with FlipChem. Being a steady-state model, it adapts to auroral precipitation, as illustrated in Figure 8 showing the $O_2^+$ density for both models. FlipChem predicts a large $O_2^+$ fraction, overestimating it above 120

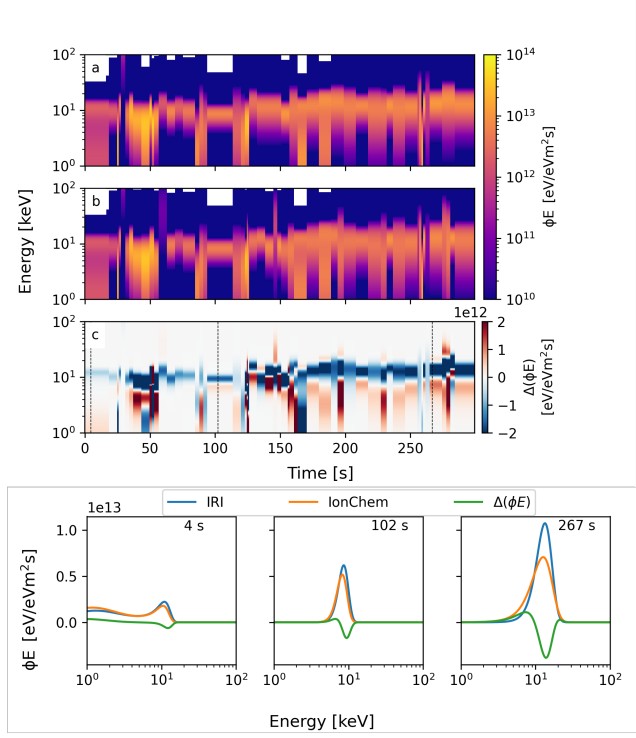

**Figure 6.** The energy flux spectrum using IRI composition (a) is compared to the one calculated with IonChem composition (b). Panel (c) shows the difference between the IRI and IonChem derived spectra $\Delta I_E = I_{E,IRI} - I_{E,IC}$. The three lowest panels show slices at different times. Those time slices are marked with a dashed black line in panel (c). The negative values at around 10 keV in (c) show that with the IonChem model, a lower flux at higher energies is necessary to reproduce the electron density measurements. The difference is significant, reaching up to 50 %.

km. Furthermore, it does not capture the short-timescale variation correctly. This can be seen during times of strong and rapidly varying precipitation, e.g. at $125$ s in Figure 8. When IonChem models a reduction in $O_2^+$ fraction, FlipChem sees an increase. Lastly, noise in the temperature measurements affects reaction rates, leading to different steady-state densities. The noise in temperature is thereby propagated into the FlipChem densities. The noise in temperature is not affecting IonChem densities as much, as IonChem integrates the continuity equation dynamically. With the high time resolution (in this case $4$ s for temperature), IonChem tends towards, but does not reach, steady-state in that interval, dampening the effect of noise in temperature. Furthermore, the noise in subsequent time bins has an equalizing effect.

The differential energy fluxes produced using FlipChem and IonChem are compared in Fig. 9. Again, a systematic shift of the flux at $10$ keV - $20$ keV to slightly lower energies can be seen at all time-steps, due to the higher $O_2^+$ fraction at heights below $100$ km in the IonChem model (see Figure 8). Furthermore, between $30$ s - $70$ s into the event, the flux at intermediate energies of $3$ keV - $9$ keV increases, due to the decreased $O_2^+$ fraction.

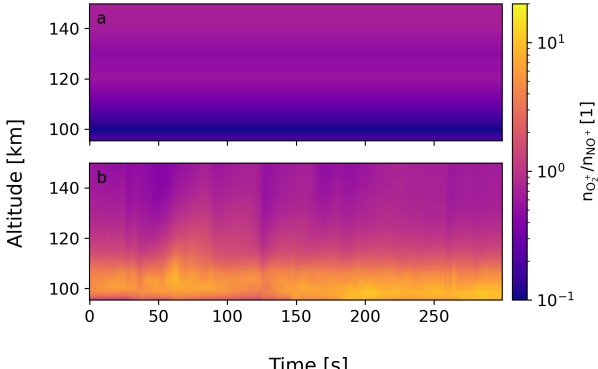

**Figure 7.** The ratio $r = n_{O_2^+}/n_{NO^+}$ is shown for (a) IRI composition and (b) IonChem composition. IonChem produces a much higher ratio at low altitudes, and also transient elevated ratios at higher altitudes. This change in $n_{O_2^+}/n_{NO^+}$ ratio implies differences in the effective recombination rate.

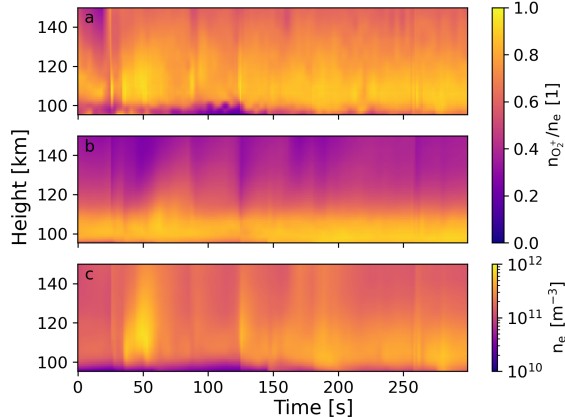

**Figure 8.** The top panel (a) shows the $O_2^+$ fraction produced by FlipChem, the middle panel (b) shows the IonChem model, and the lowest panel (c) shows the electron density for context, retrieved from the IonChem model. The noise in temperature is propagated to the fractions in the FlipChem, causing the patches in the uppermost panel.

The difference in $O_2^+$ fraction between the two models during strong precipitation has been investigated further, as the decrease seen in IonChem during intense precipitation might seem counterintuitive at first. Since $O_2^+$ is produced directly by impact ionization, one might also expect its mixing ratio to increase. Instead, we find the $O_2^+$ fraction temporarily decreasing, while $NO^+$ and $O^+$ fractions are increasing, as shown in Figure 10.

To study why the $NO^+$ fraction increases, while $O_2^+$ decreases, a simulation with constant, strong precipitation is run. IonChem is initialized with the densities found just before $125$ s in Figure 10, and the precipitation is set to what we find at

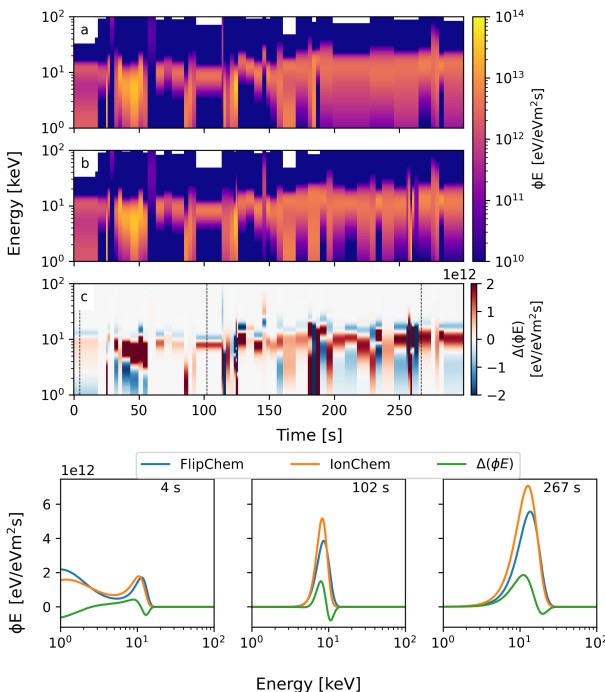

**Figure 9.** The energy flux spectrum using FlipChem composition (a) is compared to the one calculated with IonChem composition (b). Panel (c) shows the difference between the FlipChem and IonChem derived spectra $\Delta I_E = I_{E,FC} - I_{E,IC}$. The three lowest panels show a slice at different times. A systematic downshift from the highest energies (around $10\,\text{keV}$) in each timestep can be seen. Additionally, e.g., at around 50 seconds, the intermediate energies around $5\,\text{keV}$ are enhanced.

125 s. Both $O_2^+$ and $NO^+$ densities rise, as shown in Figure 11. $NO^+$ has a faster initial growth rate, which lowers the $O_2^+/NO^+$ ratio, before it recovers. Having a higher recombination rate, $NO^+$ tends to the steady-state quicker than $O_2^+$. Furthermore, as atomic oxygen becomes more abundant with height, $N_2^+$ predominantly reacts with atomic oxygen producing $NO^+$ (Ulich et al., 2000). Th reaction rate enables the $NO^+$ fraction to increase rapidly. This causes $NO^+$ to become more abundant at high altitudes. After a period of high steady ionization a steady-state with an increased $O_2^+/NO^+$ mixing ratio is approached again.

Lastly, we find a significant $O^+$ fraction down to $120\,\text{km}$ during strong precipitation, impacting the mean ion mass (Figure 10). This has consequences in fitting the electron and ion temperatures in ISR measurements, as they depend on the mean ion mass. Fitting the temperatures and mean ion mass simultaneously is difficult (e.g., Waldteufel, 1971), therefore, a model for the ion mass is commonly used. Even though the surges are short-lived, they may affect, for example, high-resolution analysis as in Tesfaw et al. (2022).

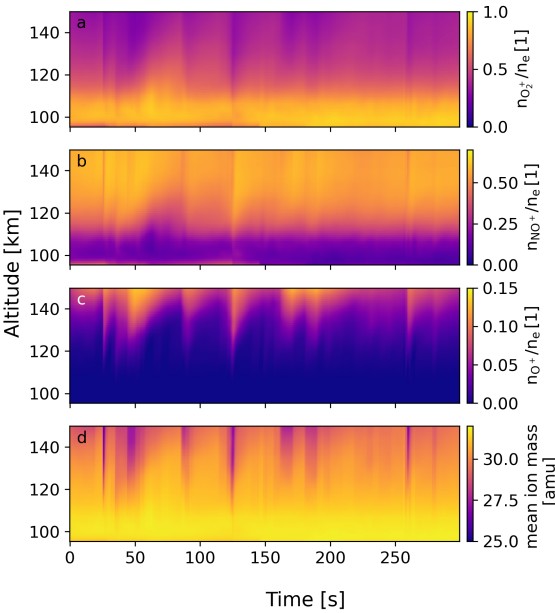

**Figure 10.** The fractions of $O_2^+$ (a), $NO^+$ (b) and $O^+$ (c) are shown. When precipitation spikes, we find enhancements in the mixing ratio of $NO^+$ and $O^+$. Furthermore, significant $O^+$ densities down to 110 km are found. This has an effect on the mean ion mass shown in (d).

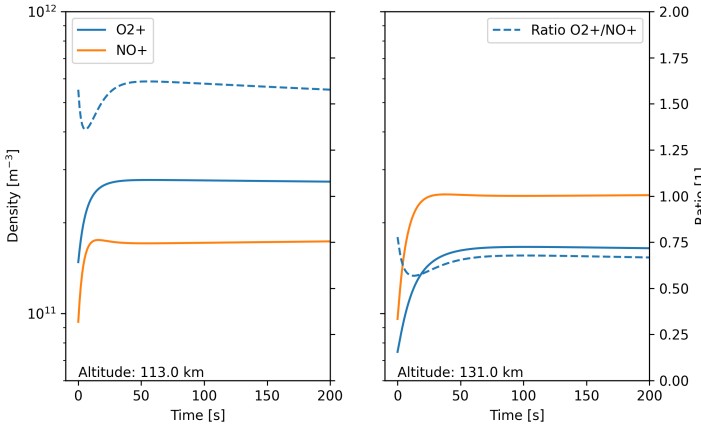

**Figure 11.** The evolution of $O_2^+$ and $NO^+$ densities during strong precipitation is shown at two altitudes. In both cases $NO^+$ tends quicker towards a steady-state, decreasing the $O_2^+/NO^+$ ratio temporarily. At lower heights, $O_2^+$ remains the dominating species.

## 4    Discussion

A new method to improve the inversion of electron density profiles to primary electron spectra is presented, that couples the fully dynamic ionospheric chemistry model IonChem with the time-dependent inversion algorithm ELSPEC. The iterative approach to simultaneously obtain electron energy spectra and ion compositions used here shows good results. It converges over a few iterations and is robust against uncertain initial conditions. The importance of using a dynamic model is shown with an example of transient effects in the $O_2^+/NO^+$ ratio and the $O^+$ fraction.

ELSPEC presents an improvement compared to steady-state inversion procedures (Vondrak and Baron, 1977; Brekke et al., 1989; Miyoshi et al., 2015; Kaeppler et al., 2015), as it integrates the electron continuity equation numerically without assuming balanced ionization and recombination rate at all times. Time-dependent inversion algorithms have been used, but assumed time-constant recombination rates (Kirkwood, 1988; Semeter and Kamalabadi, 2005). ELSPEC integrates the electron continuity equation instead of using it to explicitly calculate ionization profiles, similar to Dahlgren et al. (2011). This reduces

the impact of measurement noise. Here, ELSPEC and IonChem adjust the ion composition iteratively, while Dahlgren et al. (2011) used the Southampton ion-chemistry model to calculate the ion densities once, corresponding to one iteration. A similar method is used by Turunen et al. (2016), where the Sodankylä Ion-Neutral Chemistry model (SIC) is employed. The best-fitting ionization profile is found iteratively by integrating the coupled continuity equations of electrons and ions. The energy spectrum is then calculated using the CARD method, assuming steady-state conditions.

The method presented here is a tool that can help to improve studies of auroral precipitation, such as Tesfaw et al. (2023), and will be of interest for research with EISCAT_3D radar (McCrea et al., 2015). ELSPEC can be used to provide additional insight in conjunction with other observations, complementing the field-aligned ISR measurements, for example, with optical measurements in the horizontal direction (Wedlund et al., 2013; Tuttle et al., 2014) or satellite conjunctions (Kirkwood and Eliasson, 1990).

Further improvements to the IonChem model could be made by taking production of excited states and spontaneous emissions into account; photoionization can be added to make it compatible with daylight conditions. With EISCAT_3Ds capability for volumetric measurements of both electron densities and plasma drifts, it should be possible to account for convection. Diffusion of neutral and ion species can be taken into account, in particular to account for transport of NO (Bailey et al., 2002). Plasmaline measurements already allow for more precise determination of electron density profiles today, see Vierinen et al.

(2016). They should become routinely available with the higher sensitivity of EISCAT_3D.

## 5    Conclusion

Improved estimates of electron energy spectra from electron density profiles can be made by combining a dynamical ion chemistry model (IonChem) with the ELSPEC algorithm. The improvement of the dynamical chemistry is that it captures the variation of the ion composition during periods of rapidly changing precipitation. We find a systematic reduction of the electron

fluxes at higher energies, in our test case at around $10\,keV$, resulting from a dynamical variation of the ion composition. The cause of this reduction is due to the variability of the $O_2^+$ to $NO^+$ mixing ratio. We have also made a comparison between

our dynamical chemistry and a steady-state chemistry, FlipChem. The results show that variations in ion composition during rapid variations of ionization captured by the dynamical chemistry are not captured by a steady-state model. A rapid increase of ionization rate leads to a faster transient response of $NO^+$ compared to $O_2^+$ density. Also here, a systematic reduction of the high energy tail is found when using a dynamic model, albeit to a lesser extend. Overall, modeling the ionospheric composition improves the quality of the inversion from electron density profiles to energy spectra.

*Code and data availability.* IonChem is an open-source software, that is easily extended with more reactions. All code is available on Github: https://github.com/ostald/juliaIC.

ELSPEC is available on GitHub: https://github.com/ilkkavir/ELSPEC. The robust statistics fork used here can be found on https://github.com/ostald/ELSPEC/tree/recursive.

EISCAT data supporting this research can be found in the EISCAT archives: https://portal.eiscat.se/schedule

# Appendix A: Chemical reactions

| Reaction | Rate $[\mathrm{m^3 s^{-1}}]$ | | Branching ratio | Source |
|---|---|---|---|---|
| $\mathrm{N^+ + NO \rightarrow N_2^+ + O}$ | $8.33 \times 10^{-17}(300/T_i)^{0.24}$ | | | Richards |
| $\mathrm{N^+ + NO \rightarrow NO^+ + N(4S)}$ | $4.72 \times 10^{-16}(300/T_i)^{0.24}$ | | | Richards |
| $\mathrm{N^+ + O \rightarrow O^+(4S) + N(4S)}$ | $2.2 \times 10^{-18}$ | | | Richards |
| $\mathrm{N^+ + O_2 \rightarrow NO^+ + O(1D)}$ | $0.36\begin{cases} 5.5 \times 10^{-16}(T_i/300)^{0.45} & T_i \leq 1000 \\ 9.5 \times 10^{-16} & T_i > 1000 \end{cases}$ | | | Richards |
| $\mathrm{N^+ + O_2 \rightarrow NO^+ + O}$ | $0.09\begin{cases} 5.5 \times 10^{-16}(T_i/300)^{0.45} & T_i \leq 1000 \\ 9.5 \times 10^{-16} & T_i > 1000 \end{cases}$ | | | Richards |
| $\mathrm{N^+ + O_2 \rightarrow O^+(4S) + NO}$ | $0.05\begin{cases} 5.5 \times 10^{-16}(T_i/300)^{0.45} & T_i \leq 1000 \\ 9.5 \times 10^{-16} & T_i > 1000 \end{cases}$ | | | Richards |
| $\mathrm{N^+ + O_2 \rightarrow O_2^+ + N(2D)}$ | $0.15\begin{cases} 5.5 \times 10^{-16}(T_i/300)^{0.45} & T_i \leq 1000 \\ 9.5 \times 10^{-16} & T_i > 1000 \end{cases}$ | | | Richards |
| $\mathrm{N^+ + O_2 \rightarrow O_2^+ + N(4S)}$ | $0.35\begin{cases} 5.5 \times 10^{-16}(T_i/300)^{0.45} & T_i \leq 1000 \\ 9.5 \times 10^{-16} & T_i > 1000 \end{cases}$ | | | Richards |
| $\mathrm{N^+ + H \rightarrow H^+ + N(4S)}$ | $3.6 \times 10^{-18}$ | | | Rees |
| $\mathrm{N_2^+ + e^- \rightarrow N(4S)}$ | $2.2 \times 10^{-13}(300/T_e)^{0.39}$ | | 2 | Richards |
| $\mathrm{N_2^+ + N(4S) \rightarrow N_2 + N^+}$ | $1.0 \times 10^{-17}$ | | | Richards |
| $\mathrm{N_2^+ + NO \rightarrow N_2 + NO^+}$ | $3.6 \times 10^{-16}$ | | | Richards |
| $\mathrm{N_2^+ + O \rightarrow NO^+ + N(4S)}$ | $1.33 \times 10^{-16}(300/T_i)^{0.44}$ | | | Richards |
| $\mathrm{N_2^+ + O \rightarrow O^+(4S) + N_2}$ | $7.0 \times 10^{-18}(300/T_i)^{0.23}$ | | | Richards |
| $\mathrm{N_2^+ + O_2 \rightarrow O_2^+ + N_2}$ | $\begin{cases} 5.1 \times 10^{-17}(300/T_i)^{1.16} & T_i \leq 1000 \\ 1.26 \times 10^{-17}(T_i/1000)^{0.67} & T_i > 1000 \end{cases}$ | | | Richards |
| $\mathrm{NO^+ + e^- \rightarrow N(2D) + O}$ | $3.4 \times 10^{-13}(300/T_e)^{0.85}$ | | | Richards |
| $\mathrm{NO^+ + e^- \rightarrow N(4S) + O}$ | $0.6 \times 10^{-13}(300/T_e)^{0.85}$ | | | Richards |
| $\mathrm{O^+(4S) + H \rightarrow O + H^+}$ | $6.4 \times 10^{-16}$ | | | Richards |
| $\mathrm{O^+(4S) + N(2D) \rightarrow N^+ + O}$ | $1.3 \times 10^{-16}$ | | | Richards |
| $\mathrm{O^+(4S) + N_2 \rightarrow NO^+ + N(4S)}$ | $\begin{cases} 1.2 \times 10^{-18}(300/T_i)^{0.45} & T_i \leq 1000 \\ 7.0 \times 10^{-19}(T_i/1000)^{2.12} & T_i > 1000 \end{cases}$ | | | Richards |

| Reaction | Rate | | Branching | Reference |
|---|---|---|---|---|
| $O^+(4S) + NO \rightarrow NO^+ + O$ | $7.0 \times 10^{-19} (300/T_i)^{-0.87}$ | | | Richard |
| $O^+(4S) + O_2 \rightarrow O_2^+ + O$ | $\begin{cases} 1.6 \times 10^{-17} (300/T_i)^{0.52} & T_i \leq 900 \\ 9.0 \times 10^{-18} (T_i/900)^{0.92} & T_i > 900 \end{cases}$ | | | Richards |
| $O_2^+ + e^- \rightarrow O$ | $\begin{cases} 1.95 \times 10^{-13} (300/T_e)^{0.70} & T_i \leq 1200 \\ 7.39 \times 10^{-14} (1200/T_e)^{0.56} & T_i > 1200 \end{cases}$ | 2 | | Richards |
| $O_2^+ + N(2D) \rightarrow NO^+ + O$ | $1.8 \times 10^{-16}$ | | | Richards |
| $O_2^+ + N(2D) \rightarrow N^+ + O_2$ | $8.65 \times 10^{-17}$ | | | Richards |
| $O_2^+ + N(2P) \rightarrow N(4S) + O_2^+$ | $2.2 \times 10^{-17}$ | | | Richards |
| $O_2^+ + N(4S) \rightarrow NO^+ + O$ | $1.0 \times 10^{-16}$ | | | Richards |
| $O_2^+ + NO \rightarrow NO^+ + O_2$ | $4.5 \times 10^{-16}$ | | | Richards |
| $O_2^+ + N_2 \rightarrow NO^+ + NO$ | $5.0 \times 10^{-22}$ | | | Rees |
| $O^+(2D) + e^- \rightarrow O^+(4S) + e^-$ | $6.03 \times 10^{-14} (300/T_e)^{0.5}$ | | | Richards |
| $O^+(2D) + N(4S) \rightarrow N^+ + O$ | $1.5 \times 10^{-16}$ | | | Richards |
| $O^+(2D) + N_2 \rightarrow N_2^+ + O$ | $1.5 \times 10^{-16} (300/T_i)^{-0.55}$ | | | Richards |
| $O^+(2D) + N_2 \rightarrow NO^+ + N(4S)$ | $2.5 \times 10^{-17}$ | | | Richards |
| $O^+(2D) + N_2 \rightarrow O^+(4S) + N_2$ | $8.0 \times 10^{-16}$ | | | Rees |
| $O^+(2D) + NO \rightarrow NO^+ + O$ | $1.2 \times 10^{-15}$ | | | Richards |
| $O^+(2D) + O \rightarrow O^+(4S) + O$ | $1.0 \times 10^{-17}$ | | | Richards |
| $O^+(2D) + O_2 \rightarrow O_2^+ + O$ | $7.0 \times 10^{-16}$ | | | Richards |
| $O^+(2P) + e^- \rightarrow O^+(2D) + e^-$ | $1.84 \times 10^{-13} (300/T_e)^{0.5}$ | | | Richards |
| $O^+(2P) + e^- \rightarrow O^+(4S) + e^-$ | $3.03 \times 10^{-14} (300/T_e)^{0.5}$ | | | Richards |
| $O^+(2P) + N_2 \rightarrow N_2^+ + O$ | $2.0 \times 10^{-16} (300/T_i)^{-0.55}$ | | | Richards |
| $O^+(2P) + N_2 \rightarrow N^+ + NO$ | $1.0 \times 10^{-16}$ | | | Rees |
| $O^+(2P) + O \rightarrow O^+(4S) + O$ | $4.0 \times 10^{-16}$ | | | Richards |
| $O^+(2P) + O_2 \rightarrow O^+(4S) + O_2$ | $1.3 \times 10^{-16}$ | | | Richards |
| $O^+(2P) + O_2 \rightarrow O_2^+ + O$ | $1.3 \times 10^{-16}$ | | | Richards |
| $O^+(2P) + N(4S) \rightarrow N^+ + O$ | $1.0 \times 10^{-16}$ | | | Rees |
| $N(4S) + O_2 \rightarrow NO + O$ | $4.4 \times 10^{-18} \exp(-3220/T_n)$ | | | Rees |
| $N(4S) + NO \rightarrow N_2 + O$ | $1.5 \times 10^{-18} T_n^{0.50}$ | | | Rees |
| $N(2D) + O_2 \rightarrow NO + O(1D) + O$ | $5.3 \times 10^{-18}$ | | 1.00, 0.10, 0.90 | Rees |
| $N(2D) + O \rightarrow N(4S) + O$ | $2.0 \times 10^{-18}$ | | | Rees |
| $N(2D) + e^- \rightarrow N(4S) + e^-$ | $5.5 \times 10^{-16} (T_e/300)^{0.5}$ | | | Rees |
| $N(2D) + NO \rightarrow N_2 + O$ | $7.0 \times 10^{-17}$ | | | Rees |
| $H^+ + O \rightarrow O^+(4S) + H$ | $(8/9)6.0 \times 10^{-16} \sqrt{(T_i + (T_n/4)/(T_n + (T_i/16)}$ | | | Rees |

Table A1: Chemical reactions in the E-region and reaction rates, as well as branching ratios for reactions with several possible products. From Richards and Voglozin (2011) and Rees (1989).

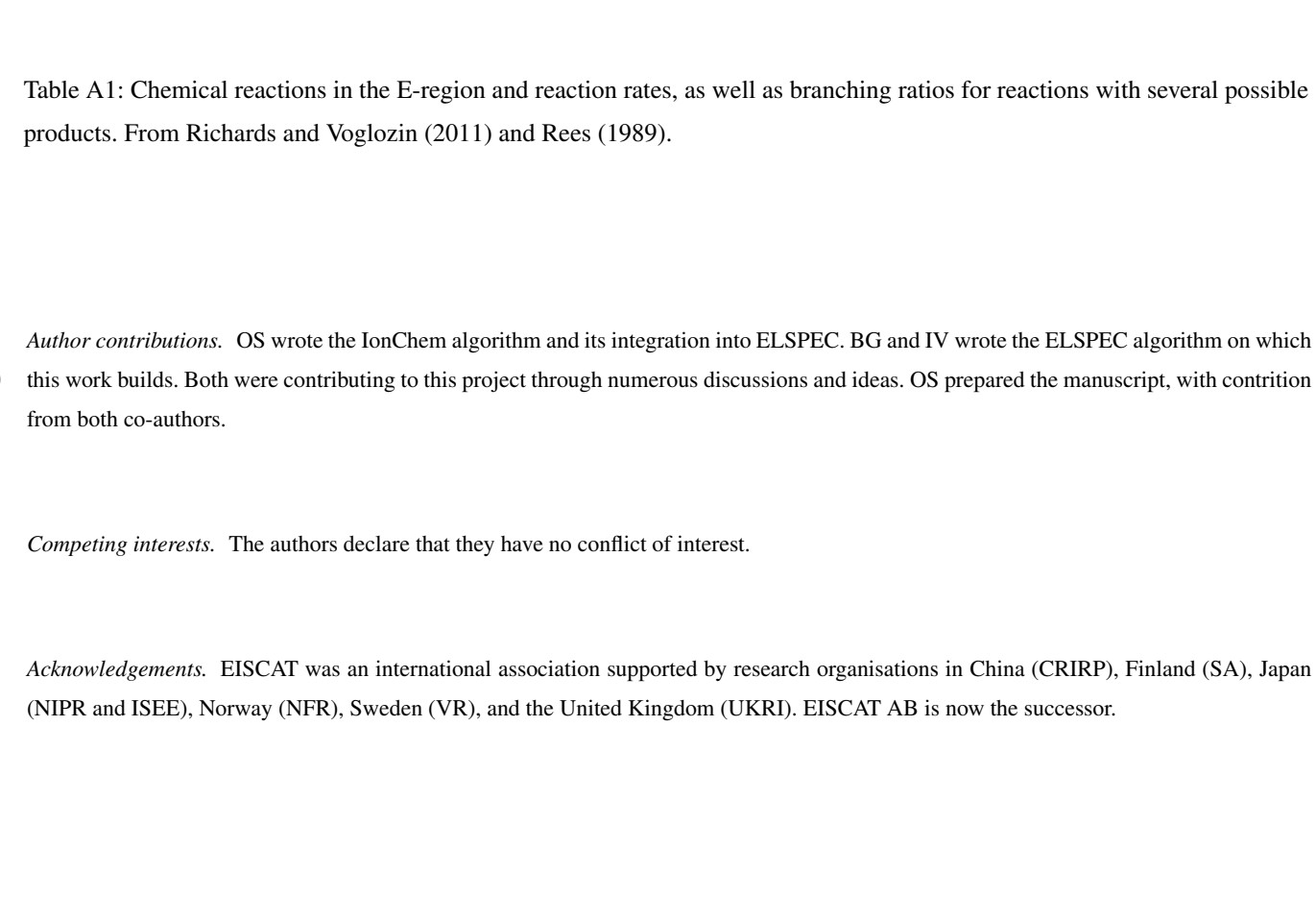

*Author contributions.* OS wrote the IonChem algorithm and its integration into ELSPEC. BG and IV wrote the ELSPEC algorithm on which this work builds. Both were contributing to this project through numerous discussions and ideas. OS prepared the manuscript, with contrition from both co-authors.

*Competing interests.* The authors declare that they have no conflict of interest.

*Acknowledgements.* EISCAT was an international association supported by research organisations in China (CRIRP), Finland (SA), Japan (NIPR and ISEE), Norway (NFR), Sweden (VR), and the United Kingdom (UKRI). EISCAT AB is now the successor.

260

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
