# Peer review of "Effect of Ionospheric Variability on the Electron Energy Spectrum estimated from Incoherent Scatter Radar Measurements"

_EGUsphere, 2025_

## Author Response (AR1)

**Author's Response**

December 2, 2025

**1 Referee 1 comments**

This manuscript presents the incorporation of time-dependent chemistry into the ELSPEC inversion technique for estimating precipitating particle information from incoherent scatter radar data. The ability to include time-dependent chemistry is a substantial improvement for ELSPEC with significant utility for future auroral research. The inverse methods are well described and clearly presented. The chemistry underlying the model has several weaknesses, and most of my critiques have to do with the details of the chemical model.

Major Comments:

1. The manuscript does not adequately explain how the concentrations of the minor neutral constituents (NO, N(4S), N(2D), H, O(1D), O(1S)) are initialized in the chemical model. Section 2.4 discusses the initial conditions for the ions in detail, but sensitivity to the initial concentration of the minor neutral constituents are not examined. I am particularly concerned about how NO is handled. The paper cites NRLMSIS2.0 (Emmert et al., 2021), which does not provide NO densities. If the more recent NRLMSIS2.1 was used instead, that version adds NO densities. See Emmert et al. 2022 Emmert, J. T., Jones, M. Jr., Siskind, D. E., Drob, D. P., Picone, J. M., Stevens, M. H., et al. (2022). NRLMSIS 2.1: An empirical model of nitric oxide incorporated into MSIS. Journal of Geophysical Research: Space Physics, 127, e2022JA030896. https://doi.org/10.1029/2022JA030896

   **Author's response**
   We thank the referee for their thoughts and inputs. Some revisions were done on that basis, attempting to increase the quality of the paper:

   The initialization of minor neutral constituents is clarified and the NRLMSIS2.1 model is used to include NO. The sensitivity to uncertain initial NO density is tested. Minor neutral constituents not represented in NRLMSIS2.1 are initialized with 0 density, with a leading time of half an hour to let them approach steady-state densities. This may not be long enough for all species to reach steady-state, but it also not a requirement. The aim is to construct a reasonable close model of the ionosphere, in regard of uncertain initial conditions. In the auroral region, where precipitation varies spatially and changes rapidly with time, the production of minor species will have a corresponding spatial and temporal variation. Therefore, we

cannot expect all species to be at steady-state. We tested the robustness of this approach towards uncertain initial conditions, and found that the differences are very small.

**Changes to the Manuscript**
The initialization of minor neutral constituents is clarified, see Page 4, Line 96. The evaluations are re-run based on the NRLMSIS2.1 model, to include NO. The sensitivity to uncertain initial NO density is tested, and added to the table of initial conditions, see Page 7, Table 1. Some adjustments to the text are made, see Page 6, Lines 134ff.

2. The manuscript does not demonstrate that the time-dependent chemistry is appropriate and reaching equilibrium for all the minor neutral constituents included, particularly NO and N(4S). NO and N(4S) are relatively long-lived species whose concentrations should be effected by vertical diffusion (both molecular and eddy diffusion) in addition to chemical production and loss. The evolution of NO in particular has been studied in depth, for example, by Bailey et al. (2002) and Barth (1992)

Bailey, S. M., C. A. Barth, and S. C. Solomon, A model of nitric oxide in the lower thermosphere, J. Geophys. Res., 107(A8), doi:10.1029/2001JA000258, 2002.

Barth, Charles A. (1992), Nitric oxide in the lower thermosphere, Planetary and Space Science, Volume 40, Issues 2–3, Pages 315-336, ISSN 0032-0633, https://doi.org/10.1016/0032-0633(92)90067-X.

Both of these models are 1-D vertical models describing production, loss, and vertical diffusion by molecular and eddy diffusion. NO takes days to reach equilibrium in these models. Barth (1992) describes running the model for 5 days for it to completely settle. Bailey et al. (2002) writes the following

"The lifetime of an NO molecule to chemical destruction (or e-folding time in the NO density) under illuminated conditions is 19 hours [Barth et al., 2001]. The lifetime of the NO molecule to diffusive transport is approximately one day [Barth, 1992]. Given that the solar illumination varies throughout the day, the abundance of NO at any one time is then representative of the level of solar energy deposition (solar irradiance and auroral energy) over the past day."

For the ground state of atomic nitrogen ( N(4S) ), the principal sink is the reaction

N(4S) + NO − > N2 + O

Therefore, I also expect N(4S) to be similar to NO and take days to equilibrate.

**Author's response**
Reaching equilibrium (or steady-state) is not a requirement. While we have uncertain initial conditions and only initialism densities that we have models for, the leading time is intended to mitigate gross mismatches that would otherwise lead to large corrections at the beginning of the dataset, and letting the uninitialized species reach non-zero densities. The largest corrections should occur at the start. Furthermore, both NO and N(4S)

are theorized to never reach equilibrium in the limited time between substorms. Auroral precipitation may increase NO and N(4S) concentrations considerably (Bailey et al, 2002). Lastly, due to the rapid variations in precipitation, many other species will also not be in equilibrium over the entire evaluation period. For all these reasons, we feel that our approach of giving the model a limited time to approach steady-state conditions is justified, without the need of actually being at steady-state. This ensures that the all species densities are in a reasonable range.

On the time-scales of auroral precipitation, vertical diffusion is small at E-region heights, see e.g. Turunen et al. (2009). Bernhardt et al. (2000) showed that an artificial airglow is subject to neutral diffusion in all directions, and drifts in response to neutral winds. Ions and electrons are subject to electrodynamic forces present in active aurora (e.g. Gustavsson et al. 2001, Krcelic et al. 2014). Furthermore, there are steep horizontal gradients in auroral precipitation. Therefore we see it more useful to include diffusion, drift and convection in 3 dimensions. This will be incorporated in future work, depending on EISCAT3D data availability.

**Changes to the Manuscript**
An explanation on why steady-state for all species is not a suitable initial condition is added to the manuscript, see Page 6, Lines 126ff. Furthermore, the intention on taking diffusion into account in future projects is added, see Page 14, Lines 238f.

3. The model comparisons between IonChem and FlipChem are using inconsistent reaction rates, which confuses the contrast between time-dependent effects and effects of different rates. The manuscript presents the contrast between IonChem and FlipChem as being primarily due to the time-independent assumptions in FlipChem. The reaction rates in appendix A are provided without a citation to their source. Nonetheless, they appear to be copied from Appendix 5 of the Rees (1989) textbook. FlipChem, however, uses the reaction rates from Richards and Voglozin (2011), which include updates from more recent laboratory measurements.

Richards, P. G., and Voglozin, D. (2011), Reexamination of ionospheric photochemistry, J. Geophys. Res., 116, A08307, doi:10.1029/2011JA016613.

If you compare appendix A to the table in Richards and Voglozin (2011), many of the rates do not match. The most meaningful way to compare the effects of time-dependent versus time-independent chemistry would be to use two models where the rates exactly match. This could be achieved by making a version of IonChem using the rates from Richards and Voglozin (2011).

**Author's response**
The reaction rates are updated. Future work will include more reactions, in particular decay of excited states.

**Changes to the Manuscript**
The reaction rates are updated, see the table A1 in the appendix (Pages 16 and 17). The evaluations are re-run based on the new reaction rates. This produced some minor adaptions, mostly due to O2+ not being the dominating ion fraction at higher altitudes anymore. See

References:

(a) Krcelic, P., Fear, R. C., Whiter, D., Lanchester, B., Brindley, N. (2024). Variability in the electrodynamics of the small scale auroral arc. Journal of Geophysical Research: Space Physics, 129, e2024JA032623. https://doi.org/10.1029/2024JA032623

(b) Gustavsson, B., et al. (2001), First tomographic estimate of volume distribution of HF-pump enhanced airglow emission, J. Geophys. Res., 106(A12), 29105–29123, doi:10.1029/2000JA900167.

(c) Bernhardt, P. A., M. Wong, J. D. Huba, B. G. Fejer, L. S. Wagner, J. A. Goldstein, C. A. Selcher, V. L. Frolov, and E. N. Sergeev (2000), Optical remote sensing of the thermosphere with HF pumped artificial airglow, J. Geophys. Res., 105(A5), 10657–10671, doi:10.1029/1999JA000366.

(d) Bailey, S. M., C. A. Barth, and S. C. Solomon, A model of nitric oxide in the lower thermosphere, J. Geophys. Res., 107(A8), doi:10.1029/2001JA000258, 2002.

(e) E. Turunen et al., "Impact of different energies of precipitating particles on NOx generation in the middle and upper atmosphere during geomagnetic storms," Journal of Atmospheric and Solar-Terrestrial Physics, vol. 71, no. 10, pp. 1176–1189, July 2009, doi: 10.1016/j.jastp.2008.07.005.

**2   Referee 2 comments**

Stalder et al. presents a method for including ion composition changes into the electron flux estimation from incoherent scatter radar observations of electron density. The time variation of the ion composition was taken into account, unlike several other previous investigations. A time-dependent model of ion composition was developed and validated against steady-state models, including the FLIP model. The updated inversion model was used on a previously investigated event. The main takeaway of the paper is that this new ion composition model could be useful for investigations of rapid electron particle precipitation.

The paper presents a useful methodology for including potential impacts from ion composition into the inversion of electron density to electron flux. While there is merit in including these impacts, I did not find the article to be a particularly compelling display of results that would lead one to abandon the

steady-state approximation. Figure 9 does not show much of a difference in the inverted numbers, and reading the scales in the bottom panel, the differences are of the order of 10%. Figure 10 seems to present the most compelling result of the investigation and the strongest reason why this method should be used. The impacts of compositional changes associated with precipitation shown here are quite interesting and would be difficult to address with a steady-state model.

Overall, I do not object to the paper being published; however, unless some effort is made to highlight what is truly significant about this investigation, it appears to be a modest advancement in the state of the art. I would recommend including more discussion, specifically on the impact of the results associated with Figure 10.

I have the following comments, which should be addressed:

1. Line 65, there needs to be more discussion about this regularization term for the least squares fitting in equation 3. Why was this particular regularization chosen and what does it mean? Considering that the reference is an unpublished reference, it should be explained in more detail why this regularization term is being used. How does this improvement compare to simply fitting the data using least squares?

   **Author's response**
   We thank the referee for their insightful and helpful comments and suggestions.

   The ElSpec methods are not based off of standard inverse-problem regularization techniques like Tikhonov regularization or Maximum Entropy, but instead uses Akaike Information Criteria for model-selection to handle the ill-posedness of the estimation problem and produce simple (in the Occam's razor sense) electron spectra. This term is not a regularization term, but a penalty that prevents overfitting. This corrected penalty term accurately accounts for small samples sizes, . It is this sample-size correction that distinguishes the corrected AIC, cAIC, from the standard AIC. For the detailed description see the first publication on ELSPEC (Virtanen et al., 2018), or Burnham et al. (2002): Model Selection and Multimodel Inference (2nd ed.).

   **Changes to the Manuscript**
   Added explanations on the corrected Akaike information criterion, see Page 3, Line 67.

2. Figure 4 does not seem to show any real difference. What is the takeaway point here?

   **Author's response**
   The entire objective with this figure is to show that the different initial conditions are producing almost exactly identical densities, such that there are no discernible differences in Figure 4. We will clarify this point in the final version.

   **Changes to the Manuscript**
   Added explanations on the relevance of Figure 4, see Page 6, Line 140.

3. Figure 6 and Figure 9, bottom panel, should really be plotted as percent differences in the flux. That would be a lot easier to understand. How much of a difference do we actually see here? Is this a 1% difference or a 10% difference?

**Author's response**
Percent differences will blow up when the reference spectrum goes to 0, i.e. at the high- and low-energy tail. We added time slices of the absolute fluxes and the difference, such that the relative difference can be better seen.

**Changes to the Manuscript**
Additional line plots are added, such that the difference in flux can be easily set in context with the fluxes, both for Figure 6 and Figure 9, see Page 10, Figure 6 and Page 12, Figure 9. Minor adaptions on the text, and comments on the amount of the difference are added, see Page 9, Lines 175ff.

4. As I stated, I think Figure 10 is the most interesting figure in the investigation, and perhaps that should be brought forward in terms of the significance of what is being done here.

**Author's response**
The focus of this paper is developing new methods for improving the inversion of electron density to electron spectra during aurora, incorporating the E-region chemistry. The improvement might be considered minor, but even if it is a small improvement it is a significant improvement, leading to a systematic correction of the estimated primary electron spectra. Every simplifying approximation made has to be scientifically justified, and then be cautiously kept in mind throughout its application with careful checks that the conditions of the approximation is fulfilled. We prefer to avoid this approximation and the ensuing worries that follow. As such we argue that this endeavor is worth to publish, and to our knowledge, has not been done before. Other authors have taken variable effective recombination rates into account (see e.g. McLennan et al. 2025), but not by using fully dynamic model of E-region chemistry. The mentioned GPI model (Kaeppler et al. 2022) uses a 5 species model, seems, to our understanding, to be more tailored to D-region studies, since it models the densities of positive and negative "light" and "heavy, cluster" ions. Such a model would be ill-suited to resolve the dynamics of $NO+$ and $O2+$, which determine the recombination rate in the E-region. E-region chemistry on its own, on the other hand, which is the context of Figure 10, we agree, has been done before. Therefore Figure 10 is not the focus of this paper, albeit an interesting finding.

**Changes to the Manuscript**
No changes.

5. I will just add that there are other investigations that have taken into account the time dependence of the chemistry, in particular, some of the implementations of the GPI D-region model. So, this is not a novel concept. You mentioned a few papers, notably some of Semeter's work, but there are also other papers which directly solve first order ODEs and then invert the associated ionization rate.

**Author's response**
See Point 4.

**Changes to the Manuscript**
No changes.

References:

- J. McLennan, A. Jaynes, R. Troyer, S. Kaeppler and M. Shumko, "Validating an Energy Flux Inversion Method with Satellite Data (Invited)," 2025 URSI Asia-Pacific Radio Science Meeting (AP-RASC), Sydney, Australia, 2025, pp. 1-4, doi: 10.46620/URSIAPRASC25/YWNE9813.

- S. R. Kaeppler, R. Marshall, E. R. Sanchez, D. H. Juarez Madera, R. Troyer, and A. N. Jaynes, "pyGPI5: A python D- and E-region chemistry and ionization model," Front. Astron. Space Sci., vol. 9, Dec. 2022, doi: 10.3389/fspas.2022.1028042.

- I. I. Virtanen, B. Gustavsson, A. Aikio, A. Kero, K. Asamura, and Y. Ogawa, "Electron Energy Spectrum and Auroral Power Estimation From Incoherent Scatter Radar Measurements," Journal of Geophysical Research: Space Physics, vol. 123, no. 8, pp. 6865–6887, 2018, doi: 10.1029/2018JA025636.

- Burnham, K. P., Anderson, D. R. (2002). Model Selection and Multimodel Inference (2nd ed.). New York: Springer.